# Effects of Different Hydration Strategies in Young Men during Prolonged Exercise at Elevated Ambient Temperatures on Pro-Oxidative and Antioxidant Status Markers, Muscle Damage, and Inflammatory Status

**DOI:** 10.3390/antiox12030642

**Published:** 2023-03-04

**Authors:** Tomasz Pałka, Piotr Michał Koteja, Łukasz Tota, Łukasz Rydzik, Alejandro Leiva-Arcas, Anna Kałuża, Wojciech Czarny, Tadeusz Ambroży

**Affiliations:** 1Department of Physiology and Biochemistry, Faculty of Physical Education and Sport, University of Physical Education, 31-571 Kraków, Poland; 2Institute of Sports Sciences, University of Physical Education, 31-571 Kraków, Poland; 3Faculty of Sport, UCAM, San Antonio de Murcia Catholic University Los Jerónimos Campus, 30107 Guadalupe, Spain; 4Student of the Doctoral School, University of Physical Education in Cracow, 31-571 Krakow, Poland; 5College of Medical Sciences, Institute of Physical Culture Studies, University of Rzeszow, 35-959 Rzeszow, Poland

**Keywords:** hydration, dehydration, exercise temperature, pro-oxidative and antioxidant status, muscle damage, inflammatory status

## Abstract

Physical exercise is associated with an increase in the speed of metabolic processes to supply energy to working muscles and endogenous heat production. Intense sweating caused by the work performed at high ambient temperatures is associated with a significant loss of water and electrolytes, leading to dehydration. This study aimed to examine the effectiveness of different hydration strategies in young men during prolonged exercise at elevated ambient temperatures on levels of pro-oxidative and antioxidant status, oxidative status markers (TAC/TOC), muscle cell damage (Mb, LDH), and inflammatory status (WBC, CRP, IL-1β). The study was conducted on a group of 12 healthy men with average levels of aerobic capacity. The intervention consisted of using various hydration strategies: no hydration; water; and isotonic drinks. The examination was di-vided into two main stages. The first stage was a preliminary study that included medical exami-nations, measurements of somatic indices, and exercise tests. The exercise test was performed on a cycle ergometers. Their results were used to determine individual relative loads for the main part of the experiment. In the second stage, the main study was conducted, involving three series of weekly experimental tests using a cross-over design. The change in plasma volume (∆PV) measured im-mediately and one hour after the exercise test was significantly dependent on the hydration strategy (*p* = 0.003 and *p* = 0.002, respectively). The mean values of oxidative status did not differ signifi-cantly between the hydration strategy used and the sequence in which the test was performed. Using isotonic drinks, due to the more efficient restoration of the body’s water and electrolyte balance compared to water or no hydration, most effectively protects muscle cells from the negative effects of exercise, leading to heat stress of exogenous and endogenous origin.

## 1. Introduction

Physical exercise is associated with an increase in the speed of metabolic processes to supply energy to working muscles and endogenous heat production. Due to the relatively low energy metabolism in working muscles, which does not exceed 30%, as the rate of metabolic processes increases, the amount of endogenous heat produced also rises [1,2]. During physical exercise at elevated ambient temperatures that exceed the weighted average skin temperature (about 31–32 °C), heat transfer is significantly impaired. With increasing heat stress (both exogenous and endogenous), sweat glands play a major role in maintaining homeothermia. Evaporation of water from the skin surface is the main mode of heat transfer which guarantees more efficient cooling of the body over a wider range of ambient temperatures compared to cutaneous vasodilation. The effectiveness of this thermoregulatory response is influenced by factors, such as humidity, hydration status, and the ability to mobilize water from the body’s water stores for the sweat glands [2,3,4,5].

Intense sweating caused by the work performed at high ambient temperatures is associated with a significant loss of water and electrolytes, leading to dehydration. Even a small loss of water in the order of 2% of body mass can impair exercise tolerance and negatively affect cognitive function and recovery [6,7]. The loss of electrolytes and the decrease in the volume of body fluids due to intense sweating leads to physiological stress, resulting in a greater increase in the body’s heat load [8].

During exercise under dehydrating conditions, the contribution of anaerobic glucose metabolism and glycogen utilization in energy production increases [9]. Therefore, it seems that to ensure the ability to perform prolonged exercises at high ambient temperatures, the choice of fluid supplementation should take into account not only the rate of fluid loss but also the increase in carbohydrate utilization. It has already been proven that hydration strategies based on carbohydrate–electrolyte fluid supply can significantly protect the body from dehydration and further postpone the onset of fatigue during exercise [10,11].

In addition to dehydration, performing prolonged exercise at high ambient temperatures disturbs the pro-oxidant–antioxidant balance [12]. Dehydration can translate into cellular susceptibility to oxidative status-induced damage, while the hyperthermic environment can further exacerbate it [13]. Reactive oxygen and nitrogen species, which are important components of many signaling pathways under physiological conditions, are primarily responsible for oxidative damage. However, when significantly increased, as observed during exercise and under stressful environmental conditions (i.e., heat stress), they can lead to the activation of inflammatory pathways and, ultimately, cell death [14]. Several studies have examined the effects of high temperatures on oxidative status, but they failed to take into account the hydration status of the participants [15,16].

Physical activity undertaken at high ambient temperatures and in high humidity, which impedes thermoregulatory processes leading consequently to significant dehydration, results in significant stress on muscles due to endogenous and exogenous hyperthermia. Muscle fiber damage caused by exercise and environmental conditions induces an inflammatory response associated with an increase in blood myoglobin (Mb), creatine kinase (CK), and lactate dehydrogenase (LDH) levels [17]. These proteins are among the widely used indicators of skeletal muscle damage [18]. Furthermore, post-exercise muscle damage triggers an inflammatory response associated with an increase in certain pro-inflammatory interleukins (e.g., IL-1β), which stimulate the production of C-reactive protein (CRP) in the liver. Interleukins are a larger group of polypeptides called cytokines that are directly related to inflammation associated with infiltration of neutrophils, and later macrophages, into the site of the injury [19,20,21]. Induced by intense exercise, dehydration, and hyperthermia, increased production of reactive oxygen species (ROS) further leads to pro-oxidant–antioxidant imbalance [17,22,23].

Given the importance of hydration status for the body’s exercise and recovery capabilities, it seems critical to examine the effects of different hydration strategies on disturbances in pro-oxidant–antioxidant balance, the degree of muscle cell damage, and inflammatory status. Therefore, this study aimed to examine the effectiveness of different hydration strategies in young men during prolonged exercise at elevated ambient temperatures on levels of oxidative status markers (TAC/TOC), muscle cell damage (Mb, LDH), and inflammatory status (WBC, CRP, IL- 1β).

## 2. Materials and Methods

### 2.1. Participants

The study was conducted on a group of 12 healthy men with average levels of aerobic capacity (46 mL·min·kg^−1^) according to American Heart Association standards (2020) [24,25]. We used Cochran’s Sample Size Formula with a 5% of margin of error and a confidence level of 95%; the required sample size was 12. Male participants, aged 20.67 ± 0.98 years, were characterized by body height (BH) of 177.25 ± 4.83 cm, body mass (BM) of 74.45 ± 7.6 kg, and lean body mass (LBM) of 61.18 ± 6.19 kg. The aerobic capacity (VO_2max_) of the participants was 3.68 ± 0.68 L.min, and in relative terms, 49.69 ± 6.69 mL·min·kg^−1^. The mean relative peak power (RPP) was 11.07 ± 0.78 W·kg^−1^, and in absolute values, it was 827.17 ± 97.99 W. The participants were familiarized with the experimental procedure and signed their written consent to participate in the experiment. As required by the Declaration of Helsinki, the participants were informed about the purpose of the study, the methodology used, possible side effects, and the possibility of withdrawal from the experiment at any stage without giving a reason. The experimental design was approved by the Bioethics Committee of the Regional Medical Chamber of Kraków, Poland (No. 42/KBL/OIL/2015) and was funded by the statutory research of the University of Physical Education in Krakow (GRANT 81/MN/INB/2015). The entire experiment was conducted under the supervision of certified medical personnel in accordance with current standards. During the period of the experiment, study participants did not use any stimulants, vitamins, and other supplements. A total of 12 people completed the full cycle of tests.

### 2.2. Study Design

The examination was divided into two main stages (Figure 1). The first stage was a preliminary study that included medical examinations, measurements of somatic indices, and exercise tests. Their results were used to determine individual relative loads for the main part of the experiment. In the second stage, a pivotal study was conducted, involving three series of weekly experimental tests using a cross-over design in which participants were divided into six groups of two, performing tests at 1-week intervals under three hydration conditions (I—isotonic drinks, W—water, NH—no hydration). Each week, each pair followed a different hydration strategy (Table 1).

### 2.3. Preliminary Study

In the preliminary study, the basic morphological indices of body composition were measured: body mass (BM) with an F 150S—DZA electronic balance (Sartorius, Göttingen, Germany) with an accuracy of 1 g. To assess percent body fat (PBF), fat mass (FM), and lean body mass (LBM), an eight-electrode electrical bioimpedance technique was employed, using the IOI-353 body composition analyzer (Jawon Medical, Seoul, Republic of Korea). The body height (BH) of the men studied was measured using a Martina anthropometer (Seritex, Howell, USA) with an accuracy of 0.5 cm.

The Wingate test was conducted (Bar–Or, 1987) to determine the anaerobic capacity using a Monark 875E cycle ergometer (Monark, Varberg, Sweden), during which maximal power (PP) was assessed. To determine aerobic capacity, we used a graded test to exhaustion on an ER 900 D—72,475 BIT2 cycle ergometer (Jeager, Wuppertal, Germany), which evaluated maximum oxygen uptake (VO_2max_).

The purpose of conducting the test was to establish individual ventilatory thresholds, which were used to determine the appropriate loads for the tests performed in the main part of the study. Based on the analysis of the kinetics of changes in selected indices of the respiratory system, such as VE, VE·VCO2-1, and RER, we determined the second ventilatory threshold (Wasserman et al., 1973).

The prolonged exercise was performed in a thermal chamber (120 min, at an ambient temperature of 31 ± 2 °C, relative humidity of 60 ± 3%, and airflow of less than 0.5 m·s^−1^. The physical load was constant and set at 53% ± 1%VO_2max_ and allowed for the determination of post-exercise water loss. Dehydration was determined by measuring body mass (with an accuracy of 1 g) before and after the exercise tests, taking into account the volume of urine excretion (Sartorius F 150S—DZA balance, Göttingen, Germany).

### 2.4. Main Experiments

The main experiments consisted of 3 series of tests at one-week intervals, required for the prevention of the consequences of water and electrolyte disturbances resulting from physical exercise. (Figure 1). Tests were carried out in a thermal chamber at an ambient temperature of 31 ± 2 °C, a humidity of 60 ± 3%, and airflow of less than 1 m·s^−1^ (Harvia thermo-hygrometer, Finland), Ellab electrothermometer (Ellab, Denmark), Hill’s kata-thermometer). During the test, the participants performed 120-min continuous exercise on cycle ergometers (Monark Type 834 E, Varberg, Sweden) with a load of 53 ± 1% VO_2max_. Furthermore, the cross-over experimental design (Kendall and Buckland, 1986) was employed to eliminate the effect of using individual hydration strategies on the results (Table 1). The main exercise test was preceded by a fifteen-minute acclimation to the chamber conditions, followed directly by the test.

### 2.5. Hydration Strategy

During exercise, participants consumed isotonic fluids (strategy I) with an osmolality of 270–330 mOsm/kg water, carbohydrate content in 100 mL of the drink of 6–8 g, sodium content (Na+) of 20–50 mmol/L (i.e., 460–1150 mg/L), or water (strategy W) at 120–150% of ΔBM [26]. Hydration strategies were established based on standards adopted by the American College of Sports Medicine (ACSM). According to the ACSM (1996), during exercise, drinks should have a temperature of 13–15 °C and be taken every 15–20 min with a volume of 150–300 mL. The participants performing exercise without hydration (strategy—NH) did not consume any fluids.

After completion of the 120-min exercise, the participants remained at room temperature for 90 min and consumed, as determined by the degree of dehydration recorded during the pre-test, an appropriate volume of isotonic drinks (strategy I), water (strategy W), or no fluids (strategy NH). Furthermore, the participants drank isotonic drinks or water (strategies I and W) 120 and 10 min before the test (according to ACSM standards, 1996).

### 2.6. Biochemical Measurements

Immediately before the start of the exercise test (preT) and during selected phases of recovery, i.e., immediately and 1 h, 24 h, and 48 h after exercise, venous blood was collected from the cubital vein for evaluation of biochemical and hematological indices, including markers of oxidative stress, muscle damage, and inflammatory status (Figure 2).

All blood samples were collected into EDTA (Poland) tubes by a laboratory diagnostician in accordance with current standards.

For hematological determinations, 4 mL of blood was drawn from the cubital vein. The collected material was used to determine the number of leukocytes (WBC) by an automated method using a Sysmex XN 9000 apparatus (Sysmex, Japan).

A blood volume of 6 mL was taken for biochemical determinations from the cubital vein. To separate the serum, blood samples were subjected to centrifugation (Medical Instruments Poland’s MPW 351R laboratory centrifuge). The centrifugation time was 15 min, the temperature was 4 °C, and the centrifugation speed was 1500 rpm. In serum, the following parameters were determined by enzyme-linked immunosorbent assay (ELISA) using a DRG plate reader (E-liza Mat 3000, Medical Instruments GmbH, Germany): Total Antioxidant Capacity (TAC) and Total Oxidant Capacity (TOC) of plasma/serum oxidative status detection limit 130 μmol L^−1^), myoglobin concentration (test sensitivity: 5 ng-mL^−1^), lactate dehydrogenase (detection limit: 10 U/L), and pro-inflammatory interleukin 1β (test sensitivity: 0.35 pg-mL^−1^).

To determine high-sensitivity C-reactive protein (hs-CRP), the collected material was centrifuged, labeled with the appropriate codes, and transported to an external laboratory, where it was subjected to analytical procedures. High-sensitivity CRP protein was determined from serum by immunoturbidimetric assay using a Roche Cobas instrument (Roche, England).

The changes in plasma volume (%DPV) were calculated using the Dill and Costill formula [27] modified by Harrison et al. [28]:%DPV = 100{(HBH1/HBH2)·[100 − (HCT2 · 0.874)]/[100 − (HCT1 · 0.874)] − 1}
(where HBH1 and HCT1 are the initial values of hemoglobin concentration and hematocrit, and HBH2 and HCT2 are the values of these parameters after exercise).

Post-exercise biochemical parameters were corrected for changes in the plasma volume [29]. The corrected values were calculated using the formula by Kraemer and Brown [30]:Corrected value = (%ΔPV × 0.01 × post-exercise value) + post-exercise value
where post-exercise value = Wpo, and %ΔPV is the change in plasma volume.

Hematocrit (HCT%) was measured using the micro-method with the Unipan MPW-212 centrifuge (Poland), and hemoglobin concentration (HBH g·dL^−1^) was measured using the Drabkin method. For this purpose, 5 μL of blood was mixed with 20 mL of Drabkin reagent, and the result was read on a spectrophotometer (Specol11, Medson, Poland). Hematocrit (HCT) and hemoglobin concentration (HBH) were measured using the electro-impedance and photometric analysis with the Vet-Analyser hematological device HA-22/20/from CLINDIAG SYSTEMS (Belgium).

### 2.7. Statistical Analysis

Based on the data obtained, descriptive statistics were calculated for the somatic constitution and aerobic and anaerobic capacity of the study group.

Hypotheses about the effects of hydration strategy, test sequence, and their interaction (HS, Trial, HSxTrial) on the mean values of the studied indices were tested using analysis of covariance (ANCOVA) for repeated measures. SAS 9.3 software (SAS Institute Inc., Cary, NC, USA) was used for the analyses. A generalized linear mixed model (SAS MIXED procedure) with REML estimation was used. Two models were employed for the analyses: a model that assumes equality of variance for three consecutive tests (compound symmetry) and a more complex unstructured model that often yields stronger results. The likelihood ratio (LR) test was used to compare the model fit with different variance structures. The results of the LR test for most variables provided a clear answer as to the validity of using a simpler model that takes into account fewer parameters, i.e., a model with an assumption of complex symmetry. For the variables, such as crp_preT, crp_1, Il1b_preT, Il1b_24, mb_1, mb_48, TacToc_1, and wbc_24, the results of the analysis using the unstructured model showed a better fit. Mean values were estimated via the least squares method, and then, if necessary, they were corrected via the Tukey–Kramer post-hoc test. The model that was used took into account one random variable (participant’s ID number), two independent variables (hydration strategy HS and test sequence—Trial), and their interactions.

In the absence of normality or heteroscedasticity of the values, we used the respective transformations. This concerned the variables that were in the nature of concentrations, that is, markers of muscle damage and hematological determinations. Before statistical analysis, they were transformed to a base-10 logarithmic scale. Several data items were excluded from the analysis due to errors at the time of performing the collection, errors of determination, and the effect of the outlier resulting, for example, due to an inflammatory condition diagnosed in the participant. In such cases, analyses were performed both without and with taking into account the remaining data for the respondent. If the conclusions of the two analyses were unchanged, it was decided to include such data in further analyses. Fixed effects were statistically significant at *p* < 0.05. The results of the significance tests of fixed effects are presented in the form of tables, whereas those for the marginal means are shown in figures for the sake of readability.

## 3. Results

### 3.1. Body Mass

Regardless of the hydration strategy used, the participants’ mean body mass (BM) decreased following exercise. The greatest weight loss was recorded when water was used (Table 2, Figure 3), but the differences between the groups were not statistically significant (Table 3).

### 3.2. Changes in Plasma Volume (∆PV)

The change in plasma volume (∆PV) measured immediately and one hour after the exercise test was significantly dependent on the hydration strategy (*p* = 0.003 and *p* = 0.002, respectively) (see Table 4). Immediately after exercise, there was no decrease in PV during hydration using isotonic drinks, and even a slight increase was recorded. A significant decrease in plasma volume was observed for hydration with water and the largest for no hydration. Significant differences in plasma volume were observed between the use of isotonic drinks and no hydration (*p* = 0.002), and between hydration with isotonic drinks and with water (*p* = 0.046) in both cases, consumption of an isotonic drink proved to be more beneficial. One hour after exercise, a marked increase in plasma volume was observed when isotonic drinks were used. An increase in PV was also found for hydration with water. No hydration sustained a negative plasma volume balance. It is noteworthy that one hour after exercise, significant differences in plasma volume (*p* = 0.001) were observed between no hydration and using isotonic drinks (Table 4 and Table 5).

### 3.3. Pro-Oxidant and Antioxidant Status

The mean values of pro-oxidative and antioxidant status did not differ significantly between hydration strategies and test sequences (Table 6). However, it can be noted that the values measured one hour and 24 h after the test were higher for no hydration compared to those when water or isotonic drinks were used (Figure 4, Table 7).

### 3.4. Muscle Damage

Mean myoglobin (MB) values did not differ immediately before the test or 24 and 48 h after the test between the hydration strategies. MB levels one hour after exercise were significantly lower for using isotonic drinks than for no hydration (*p* = 0.045) and, at the borderline of statistical significance, lower than for using water (*p* = 0.064) (Table 8 and Table 9).

For the test sequence, a trend can be observed in each measurement, indicating that the lowest values fall at T3, the middle values at T2, and the highest—at T1. One hour after exercise, significantly higher mean MB concentrations were recorded in T1 than in T3 (*p* = 0.002). Additionally, 24 h after the test, a significantly higher value was recorded in T1 compared to T3 (*p* = 0.001) (Figure 5, Table 9).

### 3.5. Lactate Dehydrogenase

Mean lactate dehydrogenase (LDH) levels immediately before the test, 1 h, and 24 h after the test differed significantly between hydration strategies (Table 10). At the first three measurement points, LDH concentrations were significantly higher for the use of isotonic drinks than for no hydration, and, in the pre-T and 24-h post-test measurements, significantly higher LDH concentrations were also found for the use of water compared to no hydration. In preT, LDH values were significantly higher for isotonic drink use than for no hydration (*p* = 0.003) and statistically significantly higher for water use than for no hydration (*p* < 0.001). One hour after exercise, the use of isotonic drinks resulted in significantly higher LDH concentrations compared to no hydration (*p* = 0.018). At 24 h after the test, the use of isotonic drinks resulted in significantly higher LDH concentrations than for no hydration (*p* = 0.006) and significantly higher concentrations for the use of water compared to no hydration (*p* = 0.008). The hydration strategy had no significant effect on LDH levels 48 h after exercise (Figure 6, Table 11).

### 3.6. Inflammatory Markers

The average leukocyte count (WBC) measured before the test did not differ between hydration strategies or test sequences. The hydration strategy had a significant effect on WBC measured immediately, 1 h, and 24 h after exercise. The test sequence had a significant effect on the mean values measured immediately and 24 h after exercise. Analysis of the results from the last measurement (24 h after the test) is hampered by a significant interaction of fixed factor effects (Table 12). Immediately and 1 h after exercise, higher WBC concentrations were observed for hydration with water compared to using isotonic drinks or no hydration. In the measurement immediately after exercise, there was a significantly higher WBC when using water than isotonic drinks (*p* = 0.002). Furthermore, 1 h after exercise, a significantly higher WBC concentration was recorded when using water compared to hydration using isotonic drinks (*p* = 0.002) (Table 13). Immediately after exercise, a significant difference was observed between T1 and T2 (*p* = 0.02).

In T1, 24 h post-exercise, significantly higher WBC values were recorded for using isotonic drinks compared to using water or no hydration. There was a statistically significant difference between the use of isotonic drinks and no hydration (*p* = 0.006). In T2, the highest values were recorded for no hydration. They were significantly higher than both the values recorded for using isotonic drinks (*p* = 0.006) and water (*p* < 0.001). In T3, the highest values were recorded when isotonic drinks were used, medium values when water was used, and the lowest values were obtained for no hydration. These differences were not statistically significant (Table 14).

### 3.7. Interleukin 1β

Mean interleukin 1β (IL-1β) levels differed significantly between hydration strategies only 48 h after exercise. The test sequence had a significant effect only on the mean values measured immediately before the test (preT). In other measurements, differences between mean IL-1β were not significant. One hour and 24 h after the test, it could be observed that IL-1β values were highest for no hydration. In the measurement after 24 h, IL-1β levels were significantly lower for using isotonic drinks than for water (*p* < 0.001) and significantly lower than for no hydration (*p* = 0.019) (Figure 7, Table 15 and Table 16).

### 3.8. C-Reactive Protein

The mean values of C-reactive protein (CRP) measured before the test (preT) and one hour after the test were significantly dependent on the interaction of the effects of hydration strategy and test sequence (Figure 8, Table 17).

In preT measurement in T1, there were no statistically significant differences in CRP between hydration strategies. However, it can be noted that slightly higher values were recorded for no hydration and the use of isotonic drinks, and lower values were found for using water. In T2, significantly higher CRP levels were recorded for no hydration compared to using isotonic drinks (*p* = 0.017) or water (*p* = 0.002). In T3, significantly lower CRP levels were found for no hydration compared to those with isotonic drinks (*p* = 0.007). They were also significantly lower than for using water as a hydration strategy (*p* = 0.040) (Table 18).

One hour after exercise, the highest CRP in T1 was observed for no hydration and the lowest for using water, but the differences were not significant. In T2, the highest levels were found for no hydration and the lowest for using water, with statistically significant differences (*p* < 0.001). CRP levels recorded in the absence of hydration were also significantly higher than for using isotonic drinks (*p* = 0.023). The opposite pattern was found in T3. CRP levels for using isotonic drinks were the highest and significantly higher than for no hydration (*p* = 0.007). The use of water in the test appeared to result in slightly lower CRP than the use of isotonic drinks and significantly higher than for no hydration (*p* = 0.040) (Table 18).

At 24 and 48 h after the test, no significant differences in CRP were found between hydration strategies or test sequences. However, it is worth noting that 24 h after the test, the highest CRP level was found for no hydration, whereas 48 h after the test—for the use of isotonic drinks (Table 17 and Table 19).

## 4. Discussion

The present study aimed to examine the effect of different hydration strategies used by young men during prolonged exercise at elevated ambient temperatures on oxidative status markers, muscle damage, and inflammatory status. Undertaking physical activity at elevated ambient temperatures triggers several thermoregulatory processes that involve disruption of water–electrolyte balance and body homeostasis. Understanding the effect of dehydration on the body’s recovery capacity and learning about hydration methods that would offset its negative impact will allow more effective planning of the quantity and quality of fluids given to athletes during training and sporting events conducted at high ambient temperatures to optimize the body’s potential for exhaustive exercise.

As a result of dehydration, plasma volume decreases, leading to reduced blood flow to contracting muscles, altered muscle cell metabolism, and impaired thermoregulation, especially in hot environments. Fluid restriction during exercise has been proven to hinder the performance of intense efforts in endurance tests [31,32,33,34]. Hydration has also been shown to improve subsequent exercise performance and is particularly important after prolonged exercise in hot and humid environments [35].

Colakoglu et al. [36] showed that consumption of isotonic drinks (500 mL water, 32 g carbohydrate, 120 mg calcium, 248 mg chloride, 230 mg sodium) had a significant effect on MB levels immediately and 2 h after an orienteering competition and compared to water hydration had a more pronounced protective effect on muscle fibers by preventing damage. This is consistent with our findings, which showed that prolonged exercise at elevated ambient temperatures and relatively high humidity caused significant changes in MB levels depending on the hydration strategy used. The experimental exercise caused a significant increase in MB levels as early as one hour after the test. Significantly lower values of this marker were recorded for using isotonic drinks compared to no hydration and using water. Therefore, it can be concluded that the use of isotonic drinks most effectively protected the muscle cells in the men studied for the negative effects of exercise, additionally burdened by heat stress of exogenous and endogenous origin. The use of isotonic drinks was likely to enable more efficient thermoregulation, which translated into reduced osmotic stress, muscle fiber edema, and hyperthermia.

An opposite pattern was observed for LDH, whose significantly lower levels were recorded for no hydration than for using isotonic drinks and water. Martínez-Navarro et al. [37], who compared changes in dehydration, serum electrolyte levels, and markers of muscle damage in marathon runners, found no correlation between dehydration degree and electrolyte levels after completing the race with an increase in LDH levels. Furthermore, Gutiérrez-Vargas et al. [38], who also studied the level of changes in markers of muscle damage before and after a marathon race in hot and humid conditions (temperature: 28–34 °C, humidity: 81%) despite dehydration, did not note a significant increase in LDH. Thus, it can be assumed that, unlike MB, LDH is not a reliable indicator of muscle damage in cases of exercise leading to significant dehydration.

Although the effects of dehydration on exercise-induced muscle damage remain largely unexplored, it is likely that proper hydration can effectively protect myocytes from exercise-induced heat stress of exogenous and endogenous origin.

In our study, we observed that prolonged physical exercise performed at high ambient temperatures led to increased WBC levels. Similar results were obtained by Boukelia et al. [39]; and Sabaei et al. [40]. An almost twofold increase in WBC levels measured immediately and one hour after the exercise test was observed for all hydration strategies. A significantly smaller increase in WBC was recorded for using isotonic drinks compared to water or no hydration. Similar results were obtained by Roh et al. [41], who showed that the use of isotonic drinks was more effective in protecting lymphocyte DNA from damage and reduced heat stress compared to hydration with water or no hydration. A significant increase in WBC in male athletes was also reported by Alhowikan and Al-Hazzaa [42], who studied the effect of a 10 km run, at an ambient temperature ≥ 35 °C and controlled dehydration conditions (without fluid intake). The results and reports by other researchers lead to the conclusion that prolonged exercise at high ambient temperatures causes an increase in WBC, and the magnitude of the changes may depend on the body’s hydration status. The use of isotonic drinks is associated with lower post-exercise WBC values, which is presumably related to better hydration levels, and, consequently, a more efficient thermoregulation process, less damage to muscle fibers, and, therefore, a less pronounced immune system response.

Exercise-induced muscle cell damage and heat stress trigger the immune system activation [43,44]. In our study, processes related to the activation of pro-inflammatory interleukins were initiated as a result of increased WBC levels and hyperthermia. One hour and 24 h after the test, IL−1β levels were highest for no hydration, and at 24 h, they were lowest for using isotonic drinks. However, these differences were not statistically significant. Only measurements at 48 h post-exercise showed significant differences between hydration strategies, with the lowest IL-1β found for using isotonic drinks. Similar patterns were documented by Sessions et al., 2016, who showed that fluid supplementation with carbohydrates during training at high ambient temperatures (30 °C) resulted in a faster decrease in plasma IL-1β levels compared to hydration with water.

The currently available literature lacks studies on the effects of prolonged training at high ambient temperatures and dehydration on levels of pro-inflammatory cytokines, including IL-1β. Cosio-Lima et al. [19] showed no differences in pro-inflammatory cytokine levels after prolonged cycling at high ambient temperatures (35 °C and humidity of 40%) and normal conditions (15 °C and humidity of 40%). It is worth noting, however, that study groups examined by these researchers did not differ in the degree of dehydration. The opposite conclusion was presented by Oliveira et al. [45], who compared levels of immune system activation and water–electrolyte disturbances during a marathon race in neutral and high ambient temperatures. Runners who completed the run at high temperatures showed a pronounced water–electrolyte imbalance and activation of the immune system. Therefore, it seems that dehydration can induce a body’s immune response and lead to an imbalance between pro-inflammatory and anti-inflammatory cytokines.

In the present study, the use of isotonic drinks proved to be the most effective hydration strategy in the context of inhibiting the activity of pro-inflammatory interleukins. Zembron-Lacny et al. [46] showed that IL-1β concentrations depend on training load and the degree of muscle damage (an increase in inflammatory mediators was observed with pronounced CK activity). In our study, less muscle damage and low WBC levels were observed when using isotonic drinks compared to using water and no hydration, which may have translated into lower IL-1β activity for this hydration strategy.

The activity of pro-inflammatory cytokines stimulates CRP production in the liver. In the present study, a significant difference was observed in the change in CRP levels between hydration strategies. In the first hour after exercise, they were significantly lower when using isotonic drinks or water compared to no hydration. Additionally, at 24 h after the test, a trend of positive effects of using isotonic drinks on acute phase protein activity was observed. Costello et al. [47] studied post-exercise CRP levels at high ambient temperatures under conditions of optimal hydration and controlled dehydration. It turns out that the dehydration degree did not significantly affect CRP levels. Similar observations have been made by other researchers [45,48,49,50]. In our study, due to the low increase in IL-1β and WBC levels after using isotonic drinks, there was likely no increase in CRP production for this hydration strategy. Therefore, it is impossible to state whether the hydration status affected post-exercise CRP levels.

Training in heat increases the body’s oxidative stress, but the effect of hydration status on pro-oxidant-antioxidant imbalance is still unknown. In our study, there was no effect of the hydration strategy on changes in the pro-oxidant–antioxidant balance. Slightly higher levels of oxidative status were observed in the absence of hydration than for using water or isotonic drinks, but these differences were not statistically significant. The lack of changes in the oxidative status may suggest that the use of the load of 53% VO_2max_ in the test did not induce a significant disruption of the pro-oxidant-antioxidant balance in the participants. Exercise intensity has already been shown to determine exercise-induced oxidative status and increased ROS production [51]. ROS production is likely to exceed the body’s antioxidant defense capacity and leads to damage only at exercise intensities above 60–70% of VO_2max_ and increases with rising exercise intensity [52]. It has also been shown that dehydration alone induced by passive heat stress can increase markers of oxidative status [53].

As described earlier, isotonic drinks, due to their effective restoration of the body’s water–electrolyte balance, can be more effective in preventing muscle cell damage and, consequently, reduce immune system responses. Unfortunately, the results failed to establish the effect of hydration status on pro-oxidant–antioxidant imbalance.

The present study has some limitations. Due to the small sample size, the results may be distorted. Furthermore, exercise at 53% VO_2max_ may have been too small a stimulus to induce muscle cell damage, which in turn translated into a poor immune system response and the magnitude of the pro-oxidant-antioxidant imbalance. Given the results obtained and the few reports on the subject, it is reasonable to conduct similar studies in the future on a larger number of participants and with a higher load.

### Limitations of the Study

Due to the inability to meet clear criteria for reliable measurement, the study did not use bioelectrical impedance analysis to assess body composition at the end of the experiment. Additionally, the study did not investigate ROS and RNS. These limitations should be taken into account when interpreting the results.

## 5. Conclusions

In conclusion, the use of different hydration strategies during prolonged exercise at elevated ambient temperatures provided results that may help understand the effect of dehydration status on the degree of muscle damage and inflammatory status. Using isotonic drinks, due to the more efficient restoration of the body’s water and electrolyte balance compared to water or no hydration, most effectively protects muscle cells from the negative effects of exercise leading to heat stress of exogenous and endogenous origin. Using such drinks is also associated with lower levels of inflammatory markers after experimental exercise compared to other strategies. Better hydration levels lead to less damage to muscle fibers and, consequently, a weaker response from the immune system. Prolonged physical exercise at high ambient temperatures performed at a low workload did not induce significant disturbances in the pro-oxidant–antioxidant balance, so the present study failed to determine the effect of hydration strategy on the level of oxidative status markers. Given the paucity of studies on the effects of hydration strategies on pro-oxidant–antioxidant balance, further research on this topic is warranted.

### Practical Implication

The isotonic drink demonstrated the most effective protective effect on muscle cells compared to water or no hydration during exercise in high environmental temperatures. Therefore, it is recommended to use isotonic drinks during physical exertion in high environmental temperatures.

## Figures and Tables

**Figure 1 antioxidants-12-00642-f001:**
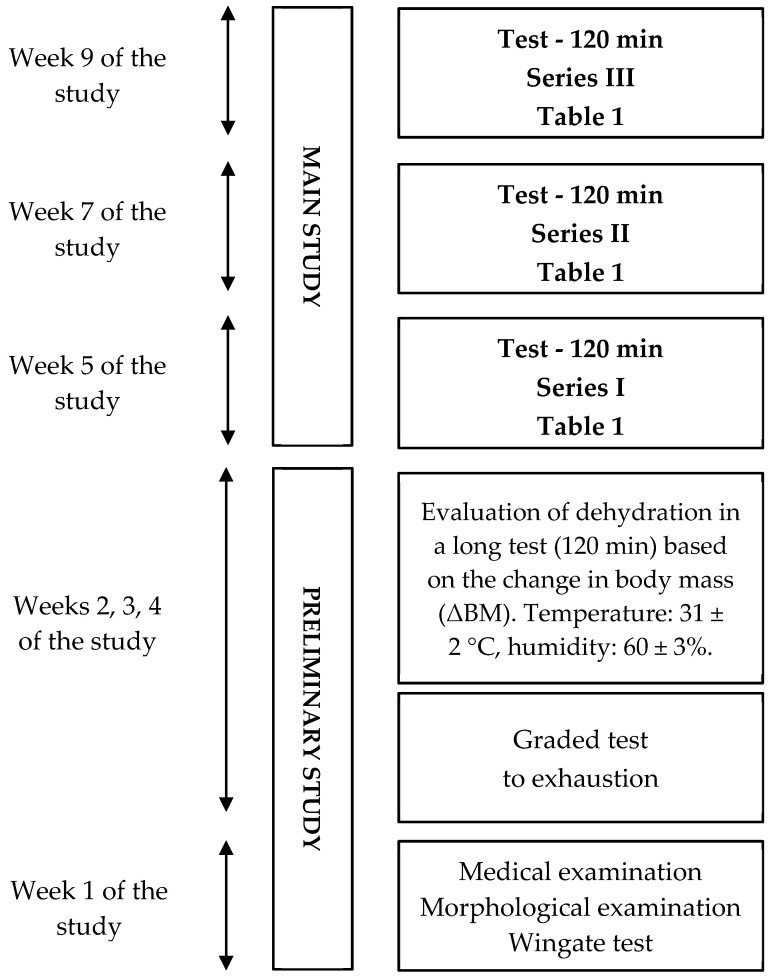
General experimental design.

**Figure 2 antioxidants-12-00642-f002:**
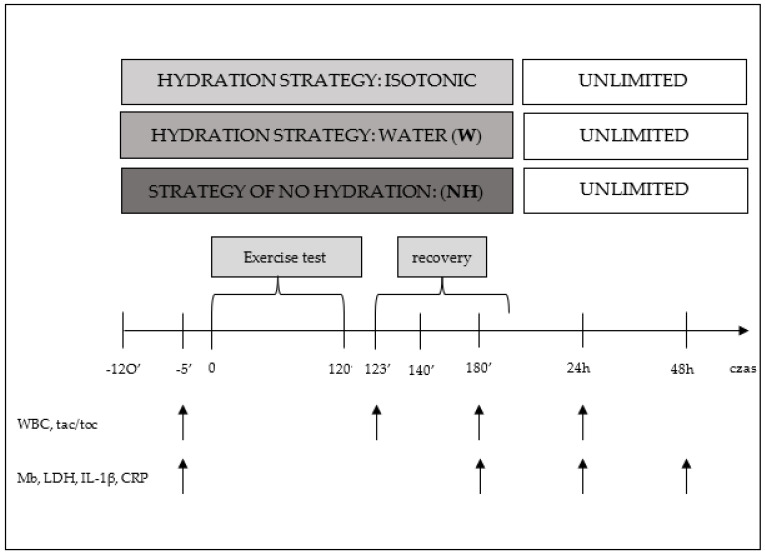
Diagram of one series of tests during the main part of the study, including measurement points WBC—white blood cells, tac/toc-pro-oxidant/antioxidant status, LDH—lactate dehydrogenase, Mb—myoglobin, IL−1β—interleukin 1β, CRP-C—reactive protein.

**Figure 3 antioxidants-12-00642-f003:**
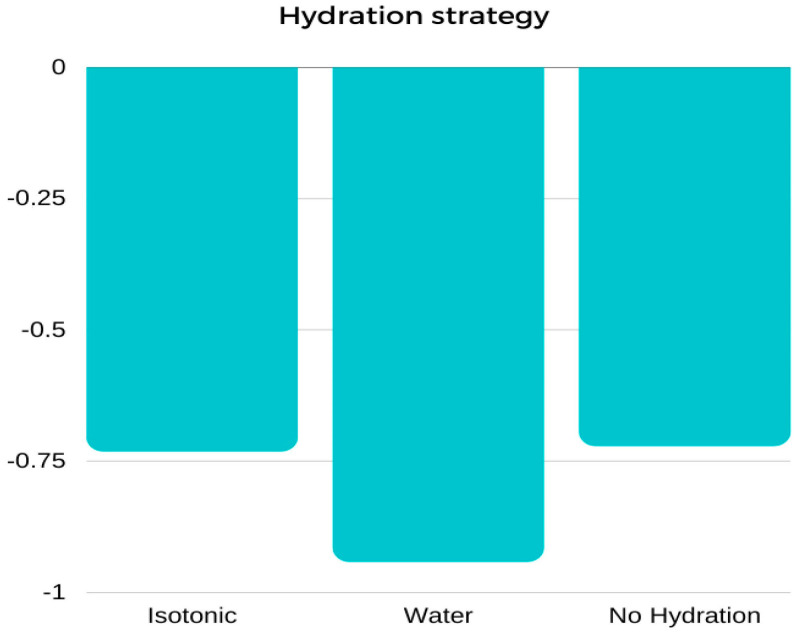
Mean values of body weight change (∆BM) [kg] between pre-test and post-test results performed using three different hydration strategies: marginal means.

**Figure 4 antioxidants-12-00642-f004:**
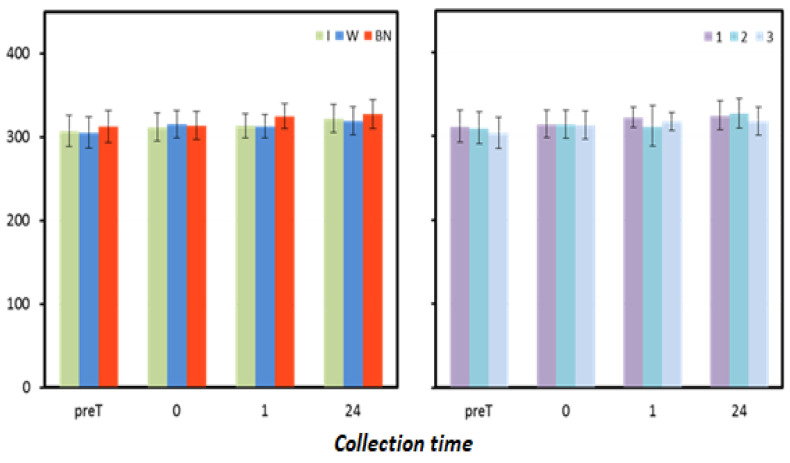
Oxidative status. Mean values estimated by the method of least squares(µmoL·L^−1^) (I—isotonic; W—water; BN—none) (immediately preT, postT, 1, 24, and 48 h) taken in 12 men in three consecutive tests (1, 2, 3).

**Figure 5 antioxidants-12-00642-f005:**
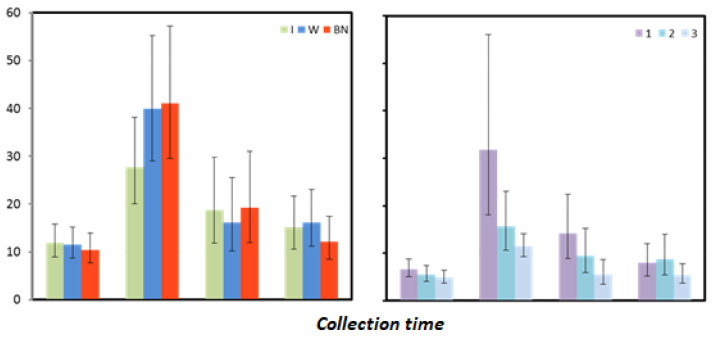
Myoglobin mean values estimated via the least squares method together with upper and lower confidence intervals for four measurements using three rehydration strategies [ng·mL^−1^].

**Figure 6 antioxidants-12-00642-f006:**
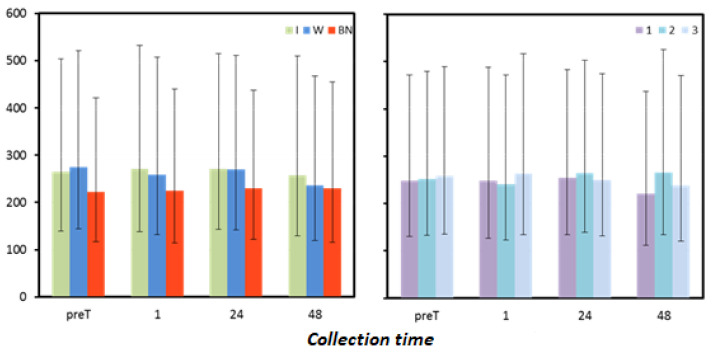
Lactate dehydrogenase mean values estimated via the least squares method with upper and lower confidence intervals for four measurements [U.mL^−1^].

**Figure 7 antioxidants-12-00642-f007:**
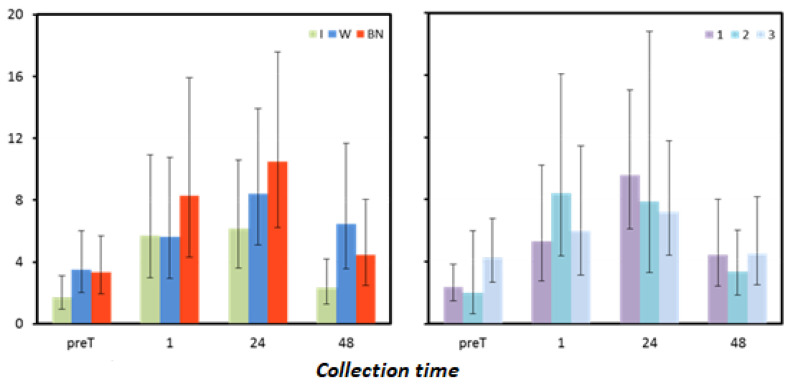
Interleukin mean values estimated via the least squares method with upper and lower confidence intervals for four measurements [pg·mL^−1^].

**Figure 8 antioxidants-12-00642-f008:**
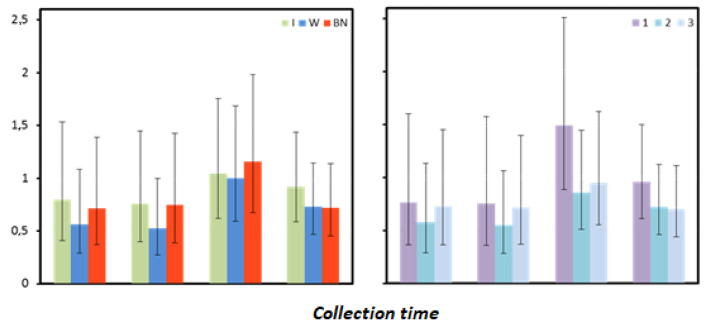
C-reactive protein mean values estimated by the least squares method with upper and lower confidence intervals for four measurements [mg·mL^−1^].

**Table 1 antioxidants-12-00642-t001:** Cross-over experimental design, in which tests are performed on six groups of two people at 1-week intervals under three hydration conditions (I—isotonic drinks, W—water, NH—no hydration), with a balanced number of interventions in each of the six possible sequences.

Group	Number of Participants	Hydration in Individual Tests		
		Test 1	Test 2	Test 3
I-W-NH	2	I	W	NH
W-NH-I	2	W	NH	I
NH-I-W	2	NH	I	W
I-NH-W	2	I	NH	W
W-I-NH	2	W	I	NH
NH-W-I	2	NH	W	I

**Table 2 antioxidants-12-00642-t002:** Mean values of the change in body mass (∆BM) [kg] between pre-test (preT) and post-test (postT) results in three consecutive tests (1, 2, 3) performed using three different hydration strategies (I, W, NH): marginal means (LSM); standard errors (SE); and lower (LCI) and upper (UCI) confidence intervals.

	Statistics	Hydration Strategy			Test Sequence		
		I	W	NH	1	2	3
∆BM	LSM	−0.73	−0.94	−0.72	−1.02	−1.01	−0.35
	SE	0.19	0.19	0.19	0.19	0.19	0.19
	LCI	−1.11	−1.32	−1.10	−1.41	−1.39	−0.74
	UCI	−0.35	−0.56	−0.33	−0.64	−0.63	0.03

**Table 3 antioxidants-12-00642-t003:** Results of analysis of variance for the change in body mass (∆BM) between the pre-test (preT) and post-test (postT) results of the exercise test: significance of (P) and F-statistic along with the number of degrees of freedom (DFi—in the numerator; DFm—in the denominator) for the fixed effects (hydration strategy, test sequence, and their interaction).

	Hydration Strategy			Test Sequence			Interaction		
		(DFi = 2)			(DFi = 2)			(DFi = 4)	
∆BM	DFm	F	P	DFm	F	P	DFm	F	P
	17.7	0.39	0.683	17.7	3.59	0.049	17.2	1.46	0.257

**Table 4 antioxidants-12-00642-t004:** Results of analysis of variance for the change in plasma volume ∆PV calculated from HB and HCT results measured before and 0, 1, and 24 h after the exercise test: significance of (P) and F-statistic along with the number of degrees of freedom (DFi—in the numerator; DFm—in the denominator) for the fixed effects (hydration strategy, test sequence, and their interaction).

Time	Hydration Strategy			Test Sequence			Interaction		
		(DFi = 2)			(DFi = 2)			(DFi = 4)	
	DFm	F	P	DFm	F	P	DFm	F	P
0	9.4	11.65	0.003	8.0	2.14	0.181	11.3	3.80	0.035
1	15.5	9.54	0.002	15.5	0.03	0.968	27.0	1.07	0.390
24	15.3	2.41	0.123	15.3	3.01	0.079	23.7	0.52	0.723

**Table 5 antioxidants-12-00642-t005:** Results for plasma volume change ∆PV [%] estimated using Dill and Costill’s formula (1974) as modified by Harisson et al. (1982). Calculations were made based on HB and HCT results measured before the test and 0, 1, and 24 h after the test, performed with different hydration strategies in three consecutive tests: marginal means (LSM), standard errors (SE), and upper (UCI) and lower (LCI) confidence intervals.

Time	Statistics	Hydration Strategy			Test Sequence		
		I	W	NH	1	2	3
0	LSM	0.5	−2.4	−4.4	−0.9	−2.2	−3.1
	SE	1.2	1.2	1.2	1.6	0.6	1.5
	LCI	−2.1	−4.9	−6.9	−4.5	−3.7	−6.6
	UCI	3.0	0.2	−1.8	2.7	−0.7	0.3
1	LSM	5.2	2.2	−1.6	2.0	2.1	1.7
	SE	1.2	1.2	1.2	1.2	1.2	1.2
	LCI	2.6	−0.3	−4.1	−0.5	−0.4	−0.8
	UCI	7.7	4.7	0.9	4.5	4.6	4.2
24	LSM	4.0	1.0	6.2	7.1	2.6	1.5
	SE	1.7	1.7	1.7	1.7	1.7	1.7
	LCI	0.5	−2.6	2.7	3.6	−0.9	−2.0
	UCI	7.6	4.5	9.8	10.6	6.1	5.1

**Table 6 antioxidants-12-00642-t006:** Results of analysis of variance for pro-oxidant/antioxidant status and oxidative status measured before the test (preT) and 0, 1, and 24 h after the exercise test: significance of (P) and F-statistic along with the number of degrees of freedom (DFi—in the numerator; DFm—in the denominator) for the fixed effects (hydration strategy, test sequence, and their interaction).

t	Hydration Strategy			Test Sequence			Interaction		
		(DFi = 2)			(DFi = 2)			(DFi = 4)	
	DFm	F	P	DFm	F	P	DFm	F	P
preT	16.7	0.33	0.723	16.7	0.39	0.683	23.8	1.45	0.249
0	17.2	0.06	0.939	17.2	0.01	0.991	26.7	2.60	0.059
1	12.7	1.04	0.381	8.0	0.43	0.663	14.1	3.62	0.031
24	17.2	0.40	0.679	17.2	0.58	0.570	25.6	1.01	0.423

**Table 7 antioxidants-12-00642-t007:** Results of pro-oxidative and antioxidant status [µmol-L−1] measured before the test (preT) and 0, 1, and 24 h after the test performed for different hydration strategies in three consecutive tests: marginal means (LSM); standard errors (SE) for values after logarithmic transformation (log); and marginal means and upper (UCI) and lower (LCI) confidence intervals restored to original form.

Collection Time	Statistic	Hydration Strategy			Test Sequence		
		I	W	NH	1	2	3
preT	LSM (log)	2.49	2.48	2.49	2.49	2.49	2.48
	SE (log)	0.01	0.01	0.01	0.01	0.01	0.01
	LSM	306.69	305.21	312.32	311.39	309.24	303.60
	LCI	288.40	287.01	293.70	292.82	290.80	285.50
	UCI	326.14	324.56	332.05	331.13	328.85	322.85
0	LSM (log)	2.49	2.50	2.50	2.50	2.50	2.50
	SE (log)	0.01	0.01	0.01	0.01	0.01	0.01
	LSM	311.82	315.21	313.62	314.12	313.62	312.82
	LCI	295.80	299.02	297.51	297.99	297.51	296.76
	UCI	328.70	332.28	330.60	331.21	330.67	329.84
1	LSM (log)	2.50	2.50	2.51	2.51	2.49	2.50
	SE (log)	0.01	0.01	0.01	0.01	0.01	0.01
	LSM	313.33	312.68	324.79	322.03	311.46	317.25
	LCI	299.43	298.74	310.38	310.10	288.00	306.48
	UCI	327.94	327.19	339.94	334.35	336.82	328.47
24	LSM (log)	2.51	2.50	2.51	2.51	2.51	2.50
	SE (log)	0.01	0.01	0.01	0.01	0.01	0.01
	LSM	321.96	319.30	327.27	324.41	326.81	317.39
	LCI	305.28	302.83	310.38	307.61	309.88	300.95
	UCI	339.47	336.67	345.06	342.06	344.59	334.66

**Table 8 antioxidants-12-00642-t008:** Results of analysis of variance for myoglobin measured before the test (preT) and 1, 24, and 48 h after the exercise test: significance of (P) and F-statistic along with the number of degrees of freedom (DFi—in the numerator; DFm—in the denominator) for the fixed effects (hydration strategy, test sequence, and their interaction).

t	Hydration Strategy			Test Sequence			Interaction		
		(DFi = 2)			(DFi = 2)			(DFi = 4)	
	DFm	F	P	DFm	F	P	DFm	F	P
preT	16.5	0.28	0.760	16.5	1.44	0.266	24.8	0.77	0.557
1	8.7	5.30	0.031	8.8	12.69	0.003	11.5	1.00	0.449
24	15.7	0.39	0.685	15.7	9.61	0.002	22.5	0.54	0.705
48	12.8	1.87	0.195	6.8	1.70	0.251	8.5	1.94	0.192

**Table 9 antioxidants-12-00642-t009:** Results for myoglobin [ng·mL^−1^] measured before the test (preT) and 1, 24, and 48 h after the test performed for different hydration strategies in three consecutive tests: marginal means (LSM); standard errors (SE) for values after logarithmic transformation (log); and marginal means and upper (UCI) and lower (LCI) confidence intervals restored to original form.

Collection Time	Statistic	Hydration Strategy			Test Sequence		
		I	W	NH	1	2	3
preT	LSM (log)	1.08	1.06	1.02	1.12	1.04	0.99
	SE (log)	0.06	0.06	0.06	0.06	0.06	0.06
	LSM	11.90	11.45	10.35	13.26	10.96	9.70
	LCI	8.97	8.64	7.69	10.00	8.14	7.31
	UCI	15.78	15.18	13.93	17.59	14.76	12.86
1	LSM (log)	1.44	1.60	1.61	1.80	1.50	1.36
	SE (log)	0.07	0.07	0.07	0.11	0.07	0.04
	LSM	27.67	40.00	41.10	63.56	31.34	22.84
	LCI	20.06	29.01	29.53	36.08	21.33	18.43
	UCI	38.16	55.17	57.21	111.97	46.05	28.31
24	LSM (log)	1.27	1.21	1.29	1.45	1.28	1.04
	SE (log)	0.10	0.10	0.10	0.10	0.10	0.10
	LSM	18.76	16.10	19.30	28.24	18.89	10.92
	LCI	11.83	10.15	11.99	17.81	11.74	6.89
	UCI	29.75	25.53	31.07	44.78	30.42	17.33
48	LSM (log)	1.18	1.21	1.08	1.20	1.24	1.03
	SE (log)	0.07	0.07	0.07	0.08	0.09	0.07
	LSM	15.09	16.06	12.15	15.85	17.34	10.71
	LCI	10.51	11.19	8.43	10.44	10.74	7.42
	UCI	21.65	23.05	17.49	24.03	28.00	15.47

**Table 10 antioxidants-12-00642-t010:** Results of analysis of variance for lactate dehydrogenase measured before the test (preT) and 1, 24, and 48 h after the exercise test: significance of (P) and F-statistic along with the number of degrees of freedom (DFi—in the numerator; DFm—in the denominator) for the fixed effects (hydration strategy, test sequence, and their interaction).

t	Hydration Strategy			Test Sequence			Interaction		
		(DFi = 2)			(DFi = 2)			(DFi = 4)	
	DFm	F	P	DFm	F	P	DFm	F	P
preT	16.0	12.73	0.001	16.0	0.33	0.723	16.1	1.45	0.263
1	16.0	5.20	0.018	16.0	1.12	0.352	16.2	0.80	0.541
24	16.0	8.51	0.003	16.0	0.91	0.421	16.1	1.58	0.229
48	16.0	2.69	0.098	16.0	6.43	0.009	16.2	1.42	0.271

**Table 11 antioxidants-12-00642-t011:** Results for lactate dehydrogenase [U·mL−1] measured before the test (preT) and 1, 24, and 48 h after the test performed for different hydration strategies in three consecutive tests: marginal means (LSM); standard errors (SE) for values after logarithmic transformation (log); and marginal means and upper (UCI) and lower (LCI) confidence intervals restored to original form.

Collection Time	Statistic	Hydration Strategy			Test Sequence		
		I	W	NH	1	2	3
preT	LSM (log)	2.42	2.44	2.35	2.39	2.40	2.41
	SE (log)	0.13	0.13	0.13	0.13	0.13	0.13
	LSM	265.34	274.22	221.72	248.26	252.23	257.57
	LCI	139.70	144.38	116.73	130.74	132.80	135.61
	UCI	503.85	520.83	421.02	471.52	479.07	489.22
1	LSM (log)	2.43	2.41	2.35	2.40	2.38	2.42
	SE (log)	0.13	0.13	0.13	0.13	0.13	0.13
	LSM	271.46	258.82	224.54	248.71	240.71	263.45
	LCI	138.39	131.98	114.50	126.82	122.74	134.34
	UCI	532.35	507.57	440.45	487.75	472.17	516.65
24	LSM (log)	2.43	2.43	2.36	2.41	2.42	2.40
	SE (log)	0.13	0.13	0.13	0.13	0.13	0.13
	LSM	271.21	269.15	230.41	254.57	264.79	249.52
	LCI	142.82	141.71	121.34	134.06	139.41	131.40
	UCI	515.11	511.09	437.52	483.39	502.81	473.91
48	LSM (log)	2.41	2.37	2.36	2.34	2.42	2.38
	SE (log)	0.14	0.14	0.14	0.14	0.14	0.14
	LSM	257.34	235.83	229.51	220.85	265.40	237.63
	LCI	129.90	119.04	115.85	111.48	133.97	119.95
	UCI	509.80	467.09	454.57	437.52	525.65	470.65

**Table 12 antioxidants-12-00642-t012:** Results of analysis of variance for WBC measured before the test (preT) and 0, 1, and 24 h after the exercise test: significance of (P) and F-statistic along with the number of degrees of freedom (DFi—in the numerator; DFm—in the denominator) for the fixed effects (hydration strategy, test sequence, and their interaction).

t	Hydration Strategy			Test Sequence			Interaction		
		(DFi = 2)			(DFi = 2)			(DFi = 4)	
	DFm	F	P	DFm	F	P	DFm	F	P
preT	16.9	2.00	0.166	16.9	1.43	0.267	23.7	2.36	0.083
0	15.7	11.81	0.001	15.7	5.80	0.013	21.3	0.83	0.520
1	16.0	9.51	0.002	16.0	2.39	0.123	24.8	0.47	0.756
24	12.6	6.13	0.014	8.3	15.64	0.002	12.2	7.69	0.003

**Table 13 antioxidants-12-00642-t013:** Results for WBC [10^3^/mm^3^] measured before the test (preT) and 0, 1, and 24 h after the test performed for different hydration strategies in three consecutive tests: marginal means (LSM); standard errors (SE) for values after logarithmic transformation (log); and marginal means and upper (UCI) and lower (LCI) confidence intervals restored to original form.

Collection Time	Statistic	Hydration Strategy			Test Sequence		
		I	W	NH	1	2	3
preT	LSM (log)	0.77	0.74	0.73	0.74	0.73	0.76
	SE (log)	0.02	0.02	0.02	0.02	0.02	0.02
	LSM	5.83	5.45	5.39	5.45	5.42	5.79
	LCI	5.33	4.97	4.92	4.98	4.95	5.29
	UCI	6.39	5.97	5.90	5.97	5.94	6.34
0	LSM (log)	0.96	1.13	1.06	1.12	1.00	1.03
	SE (log)	0.04	0.04	0.04	0.04	0.04	0.04
	LSM	9.03	13.62	11.42	13.13	9.90	10.80
	LCI	7.37	11.11	9.32	10.71	8.08	8.82
	UCI	11.07	16.68	13.98	16.08	12.13	13.24
1	LSM (log)	0.96	1.14	1.04	1.10	1.01	1.03
	SE (log)	0.04	0.04	0.04	0.04	0.04	0.04
	LSM	9.07	13.76	11.00	12.47	10.17	10.81
	LCI	7.54	11.44	9.14	10.37	8.45	8.99
	UCI	10.91	16.55	13.23	15.01	12.24	13.01
24	LSM (log)	0.81	0.74	0.77	0.80	0.72	0.80
	SE (log)	0.02	0.02	0.02	0.03	0.02	0.02
	LSM	6.48	5.50	5.84	6.29	5.28	6.27
	LCI	5.86	4.98	5.28	5.35	4.81	5.80
	UCI	7.16	6.08	6.46	7.39	5.79	6.79

**Table 14 antioxidants-12-00642-t014:** Results for WBC [10^3^/mm^3^] measured 24 h after the test performed using different hydration strategies (I, W, BN) in three consecutive tests (1, 2, 3): marginal means (LSM) and standard errors (SE) for values after logarithmic transformation (log).

WBC	Test Sequence	1			2			3		
	Strategy	I	W	NH	I	W	NH	I	W	NH
24	LSM	0.905	0.789	0.702	0.706	0.640	0.821	0.824	0.793	0.776
	SE	0.04	0.04	0.04	0.02	0.02	0.02	0.03	0.03	0.03

**Table 15 antioxidants-12-00642-t015:** Results of analysis of variance for interleukin 1β measured before the test (preT) and 1, 24, and 48 h after the exercise test: significance of (P) and F-statistic along with the number of degrees of freedom (DFi—in the numerator; DFm—in the denominator) for the fixed effects (hydration strategy, test sequence, and their interaction).

t	Hydration Strategy			Test Sequence			Interaction		
		(DFi = 2)			(DFi = 2)			(DFi = 4)	
	DFm	F	P	DFm	F	P	DFm	F	P
preT	11.6	2.78	0.103	6.6	6.46	0.028	9.3	1.34	0.327
1	15.1	0.56	0.580	15.1	0.65	0.535	23.3	1.02	0.416
24	11.8	1.98	0.182	5.9	2.53	0.162	8.1	3.09	0.081
48	14.9	5.08	0.021	14.9	0.55	0.590	22.9	0.22	0.923

**Table 16 antioxidants-12-00642-t016:** Results for interleukin 1β [pg∙mL^−1^] measured before the test (preT) and 1, 24, and 48 h after the test performed for different hydration strategies in three consecutive tests: marginal means (LSM); standard errors (SE) for values after logarithmic transformation (log); and marginal means and upper (UCI) and lower (LCI) confidence intervals restored to original form.

Collection Time	Statistic	Hydration Strategy			Test Sequence		
		I	W	NH	1	2	3
preT	LSM (log)	0.24	0.54	0.52	0.37	0.30	0.63
	SE (log)	0.12	0.11	0.11	0.09	0.20	0.09
	LSM	1.73	3.49	3.34	2.37	1.99	4.26
	LCI	0.96	2.03	1.96	1.47	0.66	2.68
	UCI	3.10	6.00	5.68	3.82	5.99	6.79
1	LSM (log)	0.76	0.75	0.92	0.73	0.92	0.78
	SE (log)	0.14	0.14	0.14	0.14	0.14	0.14
	LSM	5.71	5.62	8.30	5.32	8.39	5.97
	LCI	2.98	2.93	4.33	2.78	4.38	3.11
	UCI	10.93	10.77	15.91	10.19	16.08	11.43
24	LSM (log)	0.79	0.93	1.02	0.98	0.90	0.86
	SE (log)	0.11	0.10	0.10	0.08	0.15	0.09
	LSM	6.16	8.41	10.48	9.58	7.86	7.22
	LCI	3.59	5.09	6.25	6.10	3.28	4.43
	UCI	10.59	13.90	17.59	15.03	18.81	11.80
48	LSM (log)	0.37	0.81	0.65	0.65	0.52	0.66
	SE (log)	0.12	0.12	0.12	0.12	0.12	0.12
	LSM	2.33	6.44	4.46	4.43	3.34	4.52
	LCI	1.29	3.56	2.47	2.45	1.85	2.50
	UCI	4.21	11.65	8.07	8.01	6.04	8.18

**Table 17 antioxidants-12-00642-t017:** Results of analysis of variance for CRP measured before the test (preT) and 1, 24, and 48 h after the exercise test: significance of (P) and F-statistic along with the number of degrees of freedom (DFi—in the numerator; DFm—in the denominator) for the fixed effects (hydration strategy, test sequence, and their interaction).

t	Hydration Strategy			Test Sequence			Interaction		
		(DFi = 2)			(DFi = 2)			(DFi = 4)	
	DFm	F	P	DFm	F	P	DFm	F	P
preT	10.4	1.13	0.361	7.1	4.43	0.057	6.6	10.85	0.005
1	10.6	1.53	0.262	7.3	5.32	0.038	9.3	8.11	0.004
24	12.5	0.15	0.863	12.5	2.57	0.116	22.2	2.93	0.044
48	13.3	0.65	0.536	13.3	1.07	0.371	23.8	3.42	0.024

**Table 18 antioxidants-12-00642-t018:** Results for CRP [mg·mL^−1^] measured before (preT) and one hour (1) after the test performed using different hydration strategies (I, W, BN) in three consecutive tests (1, 2, 3): marginal means (LSM) and standard errors (SE) for values after logarithmic transformation (log).

CRP	Test Sequence	1			2			3		
	Strategy	I	W	NH	I	W	NH	I	W	NH
preT	LSM (log)	−0.097	−0.204	−0.050	−0.298	−0.491	0.065	0.094	−0.058	−0.449
	SE (log)	0.22	0.22	0.22	0.14	0.14	0.14	0.15	0.15	0.15
1	LSM (log)	−0.132	−0.214	−0.023	−0.288	−0.534	0.039	0.060	−0.095	−0.398
	SE (log)	0.22	0.22	0.22	0.14	0.14	0.14	0.14	0.14	0.15

**Table 19 antioxidants-12-00642-t019:** Results for CRP [mg∙mL^−1^] measured before the test (preT) and 1, 24, and 48 h after the test performed for different hydration strategies in three consecutive tests: marginal means (LSM); standard errors (SE) for values after logarithmic transformation (log); and marginal means and upper (UCI) and lower (LCI) confidence intervals restored to original form.

Collection Time	Statistic	Hydration Strategy			Test Sequence		
		I	W	NH	1	2	3
preT	LSM (log)	−0.10	−0.25	−0.14	−0.12	−0.24	−0.14
	SE (log)	0.13	0.13	0.13	0.14	0.13	0.14
	LSM	0.79	0.56	0.72	0.76	0.57	0.73
	LCI	0.41	0.29	0.37	0.37	0.29	0.37
	UCI	1.54	1.09	1.39	1.60	1.13	1.45
1	LSM (log)	−0.12	−0.28	−0.13	−0.12	−0.26	−0.14
	SE (log)	0.13	0.13	0.13	0.14	0.13	0.13
	LSM	0.76	0.52	0.75	0.75	0.55	0.72
	LCI	0.40	0.27	0.39	0.36	0.28	0.37
	UCI	1.45	1.00	1.43	1.57	1.06	1.39
24	LSM (log)	0.02	0.00	0.06	0.17	−0.07	−0.02
	SE (log)	0.11	0.11	0.11	0.11	0.11	0.11
	LSM	1.04	1.00	1.16	1.49	0.86	0.95
	LCI	0.62	0.60	0.68	0.89	0.51	0.55
	UCI	1.75	1.69	1.98	2.51	1.44	1.62
48	LSM (log)	−0.04	−0.14	−0.14	−0.02	−0.14	−0.16
	SE (log)	0.09	0.09	0.10	0.09	0.09	0.10
	LSM	0.92	0.73	0.72	0.96	0.72	0.70
	LCI	0.59	0.47	0.45	0.61	0.46	0.44
	UCI	1.43	1.14	1.14	1.50	1.12	1.11

## Data Availability

All data are included in the manuscript.

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
