# Peer review of "Effects of Different Hydration Strategies in Young Men during Prolonged Exercise at Elevated Ambient Temperatures on Pro-Oxidative and Antioxidant Status Markers, Muscle Damage, and Inflammatory Status"

_antioxidants, 2023, doi:10.3390/antiox12030642_

Round 1
Reviewer 1 Report
In this article, the authors examine the effectiveness of different hydration strategies in young men (n=12) during prolonged exercise at elevated ambient temperatures on levels of oxidative stress markers (TAC/ TOC), muscle cell damage (Mb, LDH), and inflammatory status (WBC, CRP, IL-1β). The results show that sing isotonic drinks, due to the more efficient restoration of the body's water and electrolyte balance, compared to water or no hydration, most 31 effectively protects muscle cells from the negative effects of exercise leading to heat stress of exogenous and endogenous origin.
1.Introduction
The authors report that …”Reactive oxygen and nitrogen species, which are important components of many signaling pathways under physiological conditions, are primarily responsible for oxidative damage”. One of the crucial points of this work is that they mention ROS and RNS, with relative oxidative damage, but they study none of it.
2. Materials and Methods
- The sample size calculation is missing, please add.
- Which kits were used for the measurements?? manufacturer name??
- Are they all ELISA or are there some colorimetrics assay??
- Have you quantified other cytokines?? i.e IL-6; IL-10??
- Have you quantified other oxidative stress biomarkers?? no biomarkers of oxidative damage appear in the text.
- Plasma Volume methods? Please insert the method.
- Hct, was measured with..??? please, indicate in the method
- As you measure by BIA, why you don’t report the data of total body water (TBW)?
- The method to determine oxidative stress markers (TAC/TOC, detection limit 130 μmol L-1) is not indicated. What is TAC and TOC please define them.
- The high-sensitivity C-reactive protein, is determinates on serum or plasma samples??
3.Results
The results are well described, but some figures/graphics are recommended to insert (i.e for table 2,4,6,8,10,12, insert a figure – histogram plot) a that can help in the immediate understanding of the results.
Please, insert table with variations of body mass and its lean and fat mass components.
Table 6 and 7 Results for oxidative stress [μmol-L-1] is uncorrect indeed you measured tac/toc- pro-oxidant/antioxidant status
Despite that, the part of oxidative stress/damage is really limited. For this reason, it would be more correct replace “oxidative stress” with Pro/Antioxidative Status, both in the text and in the title.
Discussion
Lines 471-473 “…The significant relationship between hydration strategies and LDH levels in our study may have been due to the fact that already before the exercise test, the values for this indicator differed significantly, and these differences persisted at subsequent measurement points”. The paragraph is not clear.
The significant differences recorded at basal level are related to data present in literature or reported to the three different methods of treatment reported in your study?
First: explain the differences
Two: the interpretation data may be uncorrected.
Author Response
Dear Reviewer,
Thank you very much for your time and valuable comments, which all have been considered and incorporated. The detailed list of responses is given below. We hope that the modifications and explanation will be acceptable for you.
Yours sincerely,
Rydzik, corresponding author
In this article, the authors examine the effectiveness of different hydration strategies in young men (n=12) during prolonged exercise at elevated ambient temperatures on levels of oxidative stress markers (TAC/ TOC), muscle cell damage (Mb, LDH), and inflammatory status (WBC, CRP, IL-1β). The results show that sing isotonic drinks, due to the more efficient restoration of the body's water and electrolyte balance, compared to water or no hydration, most 31 effectively protects muscle cells from the negative effects of exercise leading to heat stress of exogenous and endogenous origin.
1.Introduction
The authors report that …”Reactive oxygen and nitrogen species, which are important components of many signaling pathways under physiological conditions, are primarily responsible for oxidative damage”. One of the crucial points of this work is that they mention ROS and RNS, with relative oxidative damage, but they study none of it.
A: Dear reviewer, thank you for your attention. We would like to point out that it was an element introducing the article and explaining the physiological conditions responsible for oxidative damage and the consequences of physical exercise. We did not test ROS and RNS, information was given in limitations
- Materials and Methods
- The sample size calculation is missing, please add.
A: This has been corrected
- Which kits were used for the measurements?? manufacturer name??
A: This has been corrected
- Are they all ELISA or are there some colorimetrics assay??
A: All were ELISAs. ELISA (Enzyme Linked Immunosorbent Assay) is a commonly used immunoenzymatic method used in immunology and analytics to detect specific proteins in the tested biological material using monoclonal or polyclonal antibodies coupled with an appropriate enzyme. Each test requires coating the solid phase with a specific antibody, depending on the type of test performed. The primary antibody binding medium is 96-well polystyrene plates with a single well capacity of approximately 300-400 µL. The tested material (serum or plasma) is placed in the wells prepared in this way and, after the designated time of incubation at the appropriate temperature, removed from the wells in repeated washing processes. During incubation, the tested protein (antigen) is bound to the solid surface of the plate by a specific antibody. The next stage of the assay consists in adding a solution of detection antibody labeled with an enzyme, e.g. horseradish peroxidase, to the wells. This antibody binds to the antigen bound to the solid phase of the platelet. After washing off the unbound antibody, we add a substrate, which in the presence of an enzyme transforms into an indicator dye. The enzymatic reaction is stopped by changing the pH of the solution environment, e.g. by adding hydrochloric acid. The dye concentration is measured using an ELISA microplate reader. Based on the calibration curve, the concentration of the analyzed antigen is calculated on the basis of absorbance. DRG MedTek reagents were used for the determinations.
- Have you quantified other cytokines?? i.e IL-6; IL-10??
A: No
- Have you quantified other oxidative stress biomarkers?? no biomarkers of oxidative damage appear in the text.
A: No
- Plasma Volume methods? Please insert the method.
A: This has been corrected
- Hct, was measured with..??? please, indicate in the method
A: hematocrit number (HCT), hemoglobin concentration (HBH) determined by electro-impedance and photometric analysis using Vet-Hematology Analyzer HA-22/20/, CLINDIAG SYSTEMS (Belgium).
- As you measure by BIA, why you don’t report the data of total body water (TBW)?
A: Dehydration analysis was recorded by scale, but TBW was not recorded, added information in limitation
- The method to determine oxidative stress markers (TAC/TOC, detection limit 130 μmol L-1) is not indicated. What is TAC and TOC please define them.
A: This has been corrected
- The high-sensitivity C-reactive protein, is determinates on serum or plasma samples??
A: Highly sensitive C-reactive protein (CRP) was determined by serum immunoturbidimetry using a Cobas Roche instrument
3.Results
The results are well described, but some figures/graphics are recommended to insert (i.e for table 2,4,6,8,10,12, insert a figure – histogram plot) a that can help in the immediate understanding of the results.
A: This has been corrected
Please, insert table with variations of body mass and its lean and fat mass components.
A: Due to the inability to meet clear criteria in order to obtain the reliability of the measurement, the use of body composition analysis using the bioimpedanji method was abandoned after the end of the experiment (Dehghan and Merchant 2008).
Dehghan, M., Merchant, A.T. Is bioelectrical impedance accurate for use in large epidemiological studies?. Nutr J 7, 26 (2008). https://doi.org/10.1186/1475-2891-7-26
Table 6 and 7 àResults for oxidative stress [μmol-L-1] is uncorrect indeed you measured tac/toc- pro-oxidant/antioxidant status
A: This has been corrected
Despite that, the part of oxidative stress/damage is really limited. For this reason, it would be more correct replace “oxidative stress” with Pro/Antioxidative Status, both in the text and in the title.
A: This has been corrected
Discussion
Lines 471-473 “…The significant relationship between hydration strategies and LDH levels in our study may have been due to the fact that already before the exercise test, the values for this indicator differed significantly, and these differences persisted at subsequent measurement points”. The paragraph is not clear.
The significant differences recorded at basal level are related to data present in literature or reported to the three different methods of treatment reported in your study?
First: explain the differences
Two: the interpretation data may be uncorrected.
A: We agree with the comment, due to our misinterpretation, we decided to remove this sentence
Reviewer 2 Report
This is an interesting paper that explores a research topic that deserves greater attention. The authors need to provide a strong justification for the experimental design used. As an example, why was the exercise intensity set at 53% VO2 max and duration 120 minutes? It would also be great if some practical applications are included in the Discussion. Below are some further comments and revisions required.
Lines 21-26: the methods are vague. What was the intervention? It seems that you compared different hydration strategies, but this is not clearly described. What type of exercise was performed? Please be more specific.
Line 53: “….in the order of 2%...”
Line 100: Provide the VO2max values that are considered average for males. Additionally, use a more recent citation than 1972!
Line 104: What does RPP stand for?
Line 115: What is meant by “three cycles of the study were completed” – Please revise.
Line 143: “…evaluated maximum oxygen uptake (VO2max).”
Lines 151: The standard deviations are missing.
Line 152: Delete “…..2max)”
Line 157: What is meant by “prevent the effects of exercise” – please revise.
Line 166: Please revise the following “used in the same number of times”. It is very difficult to understand.
Line 183: “Biochemical measurements”
Line 250” “…lower (LCI) and upper (UCI)…” – please change for all tables
Lines 263-264: Changes in plasma volume in favour of which condition?
Line 523: “…were observed when using isotonic drinks…”
Line 528: “…when using isotonic drinks or water…”
Author Response
Dear Reviewer,
Thank you very much for your time and valuable comments, which all have been considered and incorporated. The detailed list of responses is given below. We hope that the modifications and explanation will be acceptable for you.
Yours sincerely,
Rydzik, corresponding author
This is an interesting paper that explores a research topic that deserves greater attention. The authors need to provide a strong justification for the experimental design used. As an example, why was the exercise intensity set at 53% VO2 max and duration 120 minutes? It would also be great if some practical applications are included in the Discussion. Below are some further comments and revisions required.
A: Thank you, everything has been corrected. Additionally, we added a practical implication
Lines 21-26: the methods are vague. What was the intervention? It seems that you compared different hydration strategies, but this is not clearly described. What type of exercise was performed? Please be more specific.
A: This has been corrected
Line 53: “….in the order of 2%...”
A: This has been corrected
Line 100: Provide the VO2max values that are considered average for males. Additionally, use a more recent citation than 1972!
A: This has been corrected
Line 104: What does RPP stand for?
A: This has been corrected
Line 115: What is meant by “three cycles of the study were completed” – Please revise.
A: This has been corrected
Line 143: “…evaluated maximum oxygen uptake (VO2max).”
A: This has been corrected
Lines 151: The standard deviations are missing.
A: This has been corrected
Line 152: Delete “…..2max)”
A: This has been corrected
Line 157: What is meant by “prevent the effects of exercise” – please revise.
A: This has been corrected
Line 166: Please revise the following “used in the same number of times”. It is very difficult to understand.
A: This has been corrected
Line 183: “Biochemical measurements”
A: This has been corrected
Line 250” “…lower (LCI) and upper (UCI)…” – please change for all tables
A: This has been corrected
Lines 263-264: Changes in plasma volume in favour of which condition?
A: This has been corrected
Line 523: “…were observed when using isotonic drinks…”
A: This has been corrected
Line 528: “…when using isotonic drinks or water…”
A: This has been corrected
Round 2
Reviewer 1 Report
The authors have addressed my points. It can be accepted in the current form.
Reviewer 2 Report
Well done on sufficiently addressing my comments and improving the quality of your manuscript.